# CONTINUAL LEARNING USING TASK CONDITIONAL NEURAL NETWORKS

## ABSTRACT

Conventional deep learning models have limited capacity in learning multiple tasks sequentially. The issue of forgetting the previously learned tasks in continual learning is known as catastrophic forgetting or interference. When the input data or the goal of learning changes, a continual model will learn and adapt to the new status. However, the model will not remember or recognise any revisits to the previous states. This causes performance reduction and re-training curves in dealing with periodic or irregularly reoccurring changes in the data or goals. Dynamic approaches, which assign new neuron resources to the upcoming tasks, are introduced to address this issue. However, most of the dynamic methods need task information about the upcoming tasks during the inference phase to activate the corresponding neurons. To address this issue, we introduce Task Conditional Neural Network which allows the model to identify the task information automatically. The proposed model can continually learn and embed new tasks into the model without losing the information about previously learned tasks. We evaluate the proposed model combined with the mixture of experts approach on the MNIST and CIFAR100 datasets and show how it significantly improves the continual learning process without requiring task information in advance.

## 1 INTRODUCTION

The human brain can adapt and learn new knowledge in response to changing environments. Hoshi et al. (1998) demonstrate that we can continually learn different tasks while retaining previously learned variations of the same or similar phenomenon and give different reactions under different contexts. Similarly, Asaad et al. (2000) have found that our neurons are task-independent. Under different context, the neurons are fired selectively with respect to the stimulus. In contrast, most of the machine learning models, in a scalable way, are not capable of adapting to changing environments quickly and automatically using artificial neurons corresponding to different tasks. Consequently, conventional machine learning models tend to forget the previously learned task after learning a new task. This scenario is known as catastrophic forgetting or interference in machine learning (McCloskey & Cohen (1989)).

Catastrophic interference problem in machine learning is one of the hurdles to implement a general artificial intelligence learning systems (Legg & Hutter (2007)). Unable to learn several tasks, a model should be trained with all the possible scenarios in advance. This requirement is intractable in practice and is not inline with the continual learning models (Thrun & Mitchell (1995)).

Continual machine learning algorithms change over time and adapt their parameters to data or learning goals changes. We refer to the learning goal or a specific part of the data with a learning goal as a *task*. The learning models are not often equipped with solutions to quickly adapt to the situations which they have seen before if their parameters have significantly changed over time by continual learning.

A variety of continual learning methods have been proposed to solve the problems mentioned above. Shin et al. (2017) proposed a memory-based approach which is to replay the trained samples to solve the forgetting problem while learning a new task. Lee et al. (2017); Kirkpatrick et al. (2017); Zeng et al. (2018) leverage the regularisation methods reduce the representational overlap of different tasks. Dynamic network approaches assign extra neuron resources to new tasks (Yoon et al. (2017)). Most of the existing solutions need to know all the task changes in advance, or this information is given

manually to the model throughout the learning. Some other works detect in-task and out-task samples to inference the task identity (Li et al. (2019); Lee et al. (2020)).

In this work, we propose a novel method to overcome the catastrophic forgetting problem. We leverage Misture of Experts (MoE) (Masoudnia & Ebrahimpour (2014)) to train the task-specific models. One of the advantage of MoE is that we can use a shallower networks and utilise the capacity as much as possible for each expert. Theoretically, the potential capacity of the combined networks is infinite. In other words, we do not need to initialise a giant network at the beginning of the training to avoid learning further tasks without forgetting them. In each expert, we integrate the probabilistic neural networks (Specht (1990)) and conventional neural networks (Haykin (1994)) to produce the task likelihood. The final prediction of each expert will be weighted by the task likelihood and calculate the task conditional probabilities. We name our approach as Task Conditional Neural Network (TCNN).

## 2 RELATED WORK

There are different approaches to address the forgetting problem in continual learning. Parisi et al. (2019) categorise these approaches into three groups: Regularisation, Memory Replay and Dynamic Network approaches.

The regularisation approaches find the overlap of the parameter space between different tasks. One of the popular algorithms in this group is Elastic Weight Consolidation (EWC) proposed by Kirkpatrick et al. (2017). EWC avoids significantly changing the parameters that are important to a learned task. It assumes the weights have Gaussian distributions and approximates the posterior distribution of the weights by the Laplace approximation. A similar idea is used in Incremental Moment Matching (IMM) proposed by Lee et al. (2017). IMM finds the overlap of the parameter distributions by smoothing the loss surface of the tasks. Zeng et al. (2018) address the forgetting problem by allowing the weights to change within the same subspace of the previously learned task. Li & Hoiem (2018) address the problem by using the knowledge distillation (Hinton et al. (2015)). They enforce the prediction of the learned tasks to be similar to the new tasks (Parisi et al. (2019)). However, these models require advance knowledge of the training tasks and the task changes.

Memory Replay methods mainly focus on interleaving the trained samples with the new tasks. A pseudo-rehearsal mechanism proposed by Robins (1995) reduce the memory requirement to store the training samples for each task. In a pseudo-rehearsal, instead of explicitly storing the entire training samples, the training samples of previously learned tasks are drawn from a probabilistic distribution model. Shin et al. (2017) propose an architecture consisting of a deep generative model and a task solver. Similarly, Kamra et al. (2017) use a variational autoencoder to generate the previously trained samples. However, this group of models are complex to train, and in real-world cases, the sampling methods do not offer an efficient solution for sporadic and rare occurrences. These models also often require advance knowledge of the change occurring.

Dynamic Networks allocate new neurons to new tasks. Yoon et al. (2017) propose Dynamic Expandable Networks (DEN) to learn new tasks with new parameters continuously. Similarly, Serrà et al. (2018) also allocate new parameters to learn new tasks. Masse et al. (2018) propose dynamic networks Context Dependent Gating (XdG) which provides the task context information to train the model. However, this group of models require the task information to be given to the model explicitly. In other words, the model knows in advance which neurons should be activated to perform each test task. To identify which neurons or experts should be used during the test state, Aljundi et al. (2017) proposed the Expert Gate and Lee et al. (2020) proposed Continual Neural Dirichlet Process Mixture (CN-DPM) leverage generative models to achieve the task-free continual learning, which does not need the task identities in the test phase. For each task, the Expert Gate and CN-DPM need to train a generator to distinguish the in-task and out-task samples. As the complexity of the dataset grows, the extra demand including parameters and computational resources of training a generator increases. Different from them, we introduce the probabilistic layer added into the discriminative model. Our method saves the memory since we do not need an extra model which is a generator to recognise the task identity. Furthermore, our approach can save the computational resources (the probabilistic layer can be simultaneously trained with classification tasks and we do not need to train a generator for each expert).

van de Ven & Tolias (2019) define different scenarios to illustrate the demand of a model for the task information. Generally, the class-incremental scenario (Class-IL) represents the model does not need the task information in advance, task-incremental scenario (Task-IL) represents the model need the task information at inference phase.

## 3    TASK CONDITIONAL NEURAL NETWORK

In the TCNN, each expert is independent of others and has the same behaviour during the training and inference phases. One model contains multiple expert corresponding to a different task. In this section, we introduce our model by taking one expert as an example.

### 3.1    ESTIMATING THE TASK LIKELIHOOD

In this work, we refer to the task likelihood as $P(t = k|x)$ to represent the sample $x$ from the $k_{th}$ task. To estimate the task likelihood efficiently, we add a probabilistic layer into the expert. Different from the conventional neural networks, this model contains two heads performing estimating task likelihood and classification. The classification head is connected to a classification layer, which is similar to a conventional neural network. The task likelihood head is connected to the probabilistic layer. The probabilistic layers contains several kernels, each of which performs the function shown in Equation 1. Where the $f(\cdot)$ is the function of hidden, $z_i$ is a vector representing the $i_{th}$ kernel, $z_i$ has the same dimension with the output from previous layer, $\Sigma$ is the covariance matrix of $z_{1:n}$, where $n$ is the number of the kernels in the probabilistic layer. Calculating the covariance matrix could be intractable due to the high dimension of $z_{1:n}$. Since the kernels can be viewed as different data patterns, we assume the kernels in the probabilistic layers are independent of each other.

$$K(f(\mathbf{x}), \mathbf{z}_i) = \exp[-\frac{1}{2}(f(\mathbf{x}) - \mathbf{z}_i)^T \Sigma^{-1}(f(\mathbf{x}) - \mathbf{z}_i)] \tag{1}$$

The Equation 1 measures the similarity between the input and existing kernels acting as anchors. The output of the probabilistic layer is an $n$ dimensional vector. Each element in the vector can be regarded as the similarity of the input to one of the existing kernel. The summation of these $n$ similarities can be viewed as the task likelihood. To estimate the task likelihood, we perform a normalised summation shown in Equation (2). The range of the task likelihood $P(t|x, \cdot)$ is from 0 to 1.

$$P(t|x, \cdot) = \frac{\sum_{i=1}^{n} K(f(\mathbf{x}), \mathbf{z}_i)}{\sum_{i=1}^{n} K(f(\mathbf{x}), \mathbf{z}_i) + 1 - \max_{j=1...n}\{K(f(\mathbf{x}), \mathbf{z}_j)\}} \tag{2}$$

While training a new expert, we jointly maximise the task likelihood and the classification likelihood by minimizing Equation (3). Where $\{x, y\}$ is the training sample set of task $k$, $C$ is the number of classes in the task $k$, $\{x^{(c)}, y^{(c)}\}$ is the samples of $c_{th}$ class, $\theta_S$ is the parameters in the hidden layers shared by classification and probabilistic layer, $\theta_D$ and $\theta_T$ are the parameters in the classification and probabilistic layers respectively, $\lambda$ is a hyper-parameter to weight the task likelihood loss, $N$ is number of samples.

$$L_v(\theta_S, \theta_D, \theta_T) = -\frac{1}{N}(\sum_{c=1}^{C} y^{(c)} \log P(y^{(c)}|x^{(c)}, \theta_D, \theta_S) + \lambda * \log P(t = k|x, \theta_T, \theta_S)) \tag{3}$$

### 3.2    TRAINING AND INFERENCE

Each expert is corresponding to a single task. We augment each samples to generate different views and make these views converge to the kernels in the probabilistic layer. A view generated by different pre-defined functions is a variation of the input. In this work, we define several functions to get a different view of the inputs. Including Zero Component Analysis (ZCA) Pal & Sudeep (2016), shift,

rotation shear and flip. More specifically, we minimize the loss function (4), where m is the number of agumented views, $L_v(\theta_S, \theta_D, \theta_T)$ is the loss for the $v_{th}$ view.

After training all $K$ experts, we augment the test data by the same functions used in training process of each expert, and estimate the $P(y|x)$ by Equation (5), where $x_v$ is the $v_{th}$ view after augmentation, $t = k$ represents the $k_{th}$ expert is responsible for the input. Overall, we do the training and inference process by averaging several different views. The idea is inspired by contrastive learning (Chen et al. (2020)). We take the in-task samples as positive pairs and out-task samples as negative pairs to train a good feature extractor. However, in the continual learning scenarios, the out-task samples is not accessible. Hence we average multiple different views of the in-task samples to learn an anchor to distinguish the in-task and out-task samples.

$$L = \frac{1}{m} \sum_{v=1}^{m} L_v(\theta_S, \theta_D, \theta_T) \tag{4}$$

$$P(y|x) \approx \frac{1}{m} \sum_{v=1}^{m} P(y_v|x_v) \approx \frac{1}{m} \sum_{k=1}^{K} \sum_{v=1}^{m} P(y_v|x_v, t=k)P(t=k|x_v) \tag{5}$$

## 4  EXPERIMENTS AND EVALUATIONS

We test our model on the Modified National Institute of Standards and Technology (MNIST) hand-written digits dataset. We also use the Canadian Institute For Advanced Research (CIFAR) 100 dataset, which is a collection of images. See Table 1. We pre-define the covariance matrix in the probabilistic layers as an identity matrix. The weight factor $\lambda$ in Equation (3) is set to 0.1. We assume each class in each task can be represented by a single data pattern, and set the number of kernels in the probabilistic layer same as the number of classes in each task.

We compare our model with several state-of-the-art approaches including memory replay approaches referring to Generative replay (GR) (Shin et al. (2017)), Brian Inspired Replay (BI-R) (van de Ven et al. (2020)) and Learning without Forgetting (LwF) (Li & Hoiem (2018)), regularisation approaches including Synaptic Intelligence (SI) (Zenke et al. (2017)) and Elastic Weight Consolidation (EWC) (Kirkpatrick et al. (2017)), dynamic networks Context Dependent Gating (XdG) (Masse et al. (2018)). Besides that, we compare our method to Expert Gate (Aljundi et al. (2017)) and CN-DPM (Lee et al. (2020)), both of which use the MoE architecture as well. We test the methods in class-incremental scenario (Class-IL) for all the methods except XdG. XdG can only perform in task-incremental scenario (Task-IL). In other words, the XdG is informed with task identity in the test phase. We also train two models as baselines. The joint represents offline training which trains a model with all the data at once. The None represents online training without any continual learning techniques.

Table 1: Dataset used in the experiments.

| Dataset | Tasks | Classes | Classes per task | Train set | Test set |
|---|---|---|---|---|---|
| MNIST | 5 | 10 | 2 | 50K | 10K |
| CIFAR100 | 20 | 100 | 5 | 50K | 10K |

To show how much the model forgets the previously learned task, we calculate the forgetting rate of task $i$ after learning task $T$. We denote $a_l^{(i)}$ as the test accuracy on the test set of task $i$ after learning task $T$ and compute the forgetting rate by Equation 6.

$$f_T^{(i)} = \max_{l \in 1, \cdot, T-1} a_l^{(i)} - a_T^{(i)} \tag{6}$$

After learning task $T$, we calculate the forgetting rate of all the previously learned tasks and show the forgetting as a lower triangular matrix. We refer this matrix as forgetting matrix to show how much the model forget the previously learned tasks during the training process.

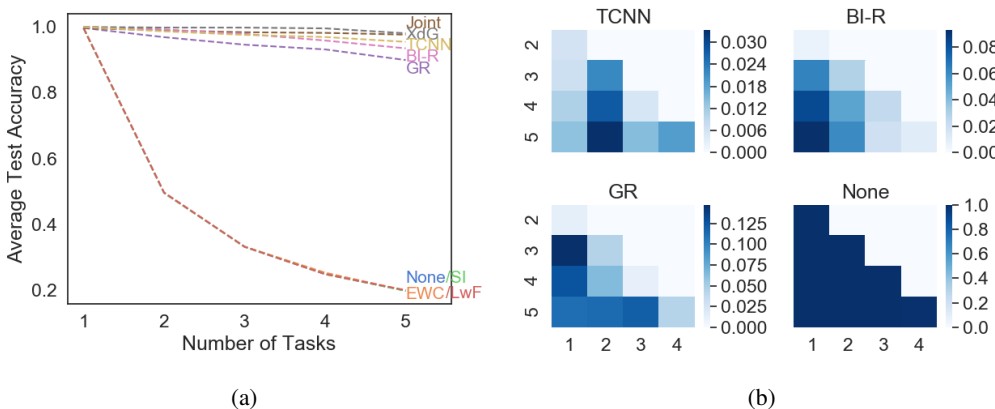

(a)                                              (b)

Figure 1: Split MNIST experiment. (a). Average test accuracy on the split MNIST experiment. All the approaches are tested in Class-IL scenario except XdG which performs Task-IL scenario. (b). Forgetting matrix of the model in the split MNIST experiment. In the forgetting matrix, the value at $i_{th}$ column and $j_{th}$ row is the forgetting rate of task $i$ after learning task $j$.

## 4.1 SPLIT MNIST EXPERIMENT

The split MNIST experiment is a benchmark experiment in continual learning field (Zenke et al. (2017); Lee et al. (2017); Masse et al. (2018)). We split the MNIST into 5 different tasks of consecutive digits referring to 0/1, 2/3, 4/5, 6/7, 8/9. The basic architecture of an expert is a convolutional neural network with two convolutional and fully-connected layers. We added a probabilistic layer containing two kernels as the extra head. We use the Adam optimizer with a learning rate 0.001.

As shown in Figure 1a, the proposed model outperforms the state-of-the-art models in this experiment and similar to a model which is trained offline and after observing all the changes. The XdG outperforms all the other methods, but it needs to be informed of the task identity during the testing state.

We also visualise the forgetting matrix to show how much the model forget the previously learned tasks, shown in Figure 1b. In the forgetting matrix, the value at $i_{th}$ column and $j_{th}$ row is the forgetting rate of task $i$ after learning task $j$. The proposed model does not forget too much learned tasks compared to other models. Notice that the forgetting rate of the second task is slightly higher than the first task in the proposed model. We analyse this scenario as the different complexity of each task has impact on the performance of continual learning methods. Similarly, Lange *et. al* demonstrate that the order of the tasks can influence the performance.

## 4.2 SPLIT CIFAR100 EXPERIMENT

In the above experiments, we only test the model to continual learn five different tasks. In this experiment, we continue to test the model in a more difficult scenario. We split the CIFAR100 datasets into 20 different tasks, each of which contains five consecutive classes. The models have to perform a 100-way classification task. Overall, the complexity of each task and the number of tasks to be learned are significantly increased compared with the split MNIST experiment. Each expert contains three convolutional layers, each of which is followed by a max pooing layer. Then we add three fully-connected layer to do the classification, and a probabilistic layer containing five kernels to produce the task likelihood. We use the Adam optimizer with a learning rate 0.001.

We have shown the result in the Figure 2a. The proposed model performs relatively well in this complex experiment and show that the proposed method can avoid the interference between the experts in the model. The XdG outperforms other methods, but it needs the task information during the test phase. We visualise the forgetting rate in the Figure 2b, which shows that the proposed model forget less of the previously learned tasks than others.

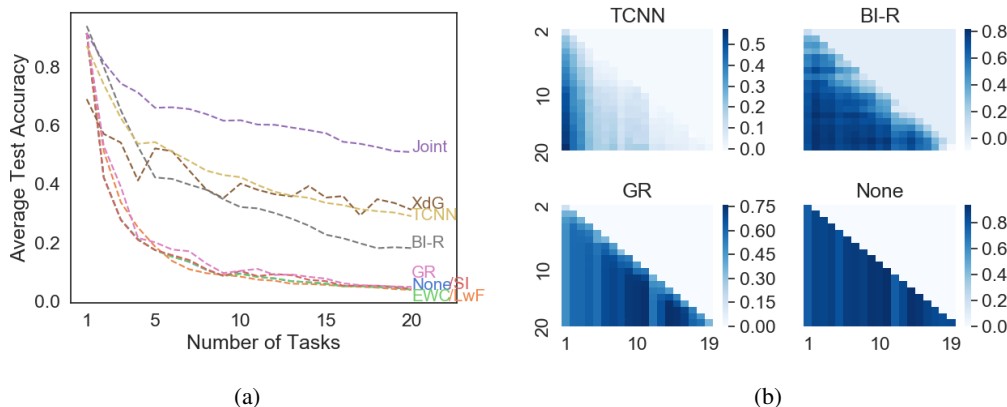

(a)                                                                  (b)

Figure 2: (a). Average test accuracy on the split CIFAR100 experiment. All the approaches are tested in Class-IL scenario except XdG which perfroms Task-IL scenario. (b). Forgetting matrix of the models in the CIFAR100 experiments. In the forgetting matrix, the value at $i_{th}$ column and $j_{th}$ row is the forgetting rate of task $i$ after learning task $j$.

| Method | Split-MNIST | Split-CIFAR100 |
|---|---|---|
| Expert Gate | 94.25 | 11.80 |
| CN-DPM | 93.81 | 20.10 |
| TCNN | 96.50 | 29.10 |

Table 2: Experimental results compared to the methods based on MoE.

For comparison, we report the accuracies of the Expert Gate, CN-DPM and TCNN in the Table 2. TCNN outperforms the Expert Gate and CN-DPM without generative models.

## 5   DISCUSSION

In this section, we take the split MNIST experiment as an example to analyse the overall performance of TCNN. We report the performance of TCNN under unknown task settings and visualise the density approximated by the probabilistic layer. Then we analyse the limitation of the model and provide some possible solutions and preliminary results.

### 5.1   REVISITING THE LEARNED TASKS AND LEARN NEW TASKS

We analyse TCNN under the task unknown setup. The task-unknown settings represent conditions in which we do not provide the task information to the model at any given time (including training phase). In other words, the model sequentially learns different tasks without being informed of the task boundaries in the training state. The model turns into test mode after it is converged on the current task, and turns into training mode if it is unconfident with the encountering task. We assume the tasks are coming in sequence. To further analyse the performance of the model, we test the model on the previously learned tasks after learning a new one. The tasks are coming in the following manner: the training set of Task 1, the test set of Task 1, the training set of Task 2, the test set of Task 1 and 2 . . . . Note that the model may add expert unnecessarily if it is unconfident with learned tasks, or fail to create new expert if it is confident with unlearned task.

To detect the changes, we set a threshold which is the $mean - std$ of the task likelihood of the model on the current training samples. If the task likelihood on the test samples is less than the threshold, the change is detected and start to learn a new expert.

We also conduct an ablation study to compare the model with and without augmentation. As shown in Figure 3, while the model is tested on the learned task (green blocks), the task likelihood remains on a higher-level. While a new task transpires, the task likelihood decreases significantly (red blocks). The

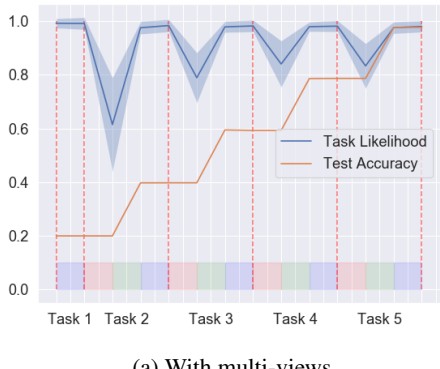 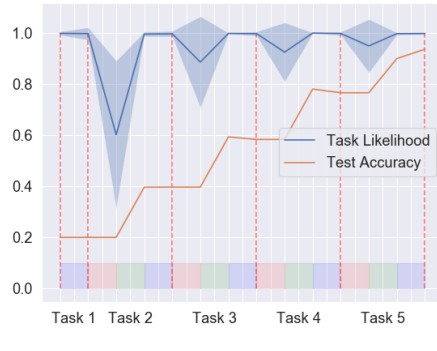

(a) With multi-views            (b) Without multi-views

Figure 3: MNIST experiment with and without multi-views augmentation. TCNN learns five different tasks sequentially without having the task information during both training and testing state. The blocks at the bottom represents different state. Blue blocks: The model is tested on the learned tasks. Green blocks: The model is learning the current task. Red blocks: The model is tested on the unseen tasks. The red dashed lines are task boundaries. The blue shadows represent the area of $mean(tlh) \pm std(tlh)$, where $tlh$ is the highest task likelihood generated by the model for the given input. When a change is detected, TCNN adapts to the new task automatically without forgetting the previous ones.

average test accuracy shows that TCNN detects the task changes and adapt to the new tasks quickly without forgetting the previously learned ones. Compared to the model with multi-view augmentation, the model without augmentation perform well on detecting new tasks in the task stream. However, the test accuracy is not satisfied. Overall, TCNN provides a unique and novel feature by automatically detecting and adapting to new tasks in an efficient way.

## 5.2 ANALYSING THE ESTIMATED DENSITY

In this section, we continue to use the split MNIST experiment to visualise the density estimated by the probabilistic layer.

As we mentioned in section 3.1, the kernels in the probabilistic layer act as anchors to distinguish the in-task and out-task samples during the test phase. The hidden layers can be viewed as a feature extractor to map the inputs to the corresponding kernels. If the expert observes the input beforehand, the features extracted by the hidden layers should match the kernels in the probabilistic layer. In contrast, the extracted features cannot match the existing kernels in the probabilistic layer if the expert has not seen the input beforehand. To verify this, we take the first expert in the split MNIST experiment and visualise the mean of the sample distribution, which is extracted by the hidden layers of the first expert, from the first and second tasks. To visualise the overall distribution, we assume the kernels in the first expert and the features extracted of the two classes follows bi-variate Gaussian distributions. The results are shown in Figure 4. The distribution of the feature distributions extracted from the samples from task 1 and the kernels from expert 1 are overlapped. In contrast, the distribution of the feature distribution extracted from the samples from task 2 is significantly different from the kernels in expert 1. Based on the Equation 1, the task likelihood will be high if the kernel can describe the extracted sample distributions perfectly, e.g. Figure 4a. Alternatively, the task likelihood will be decreased if the kernels cannot describe the extracted sample distribution, e.g. Figure 4b.

## 5.3 LIMITATIONS AND POTENTIAL SOLUTIONS

Based on the MoE framework, the demand of neuron resources increase linearly with respect to the number of learned tasks. However, each expert in the TCNN can be a shallow neural network and

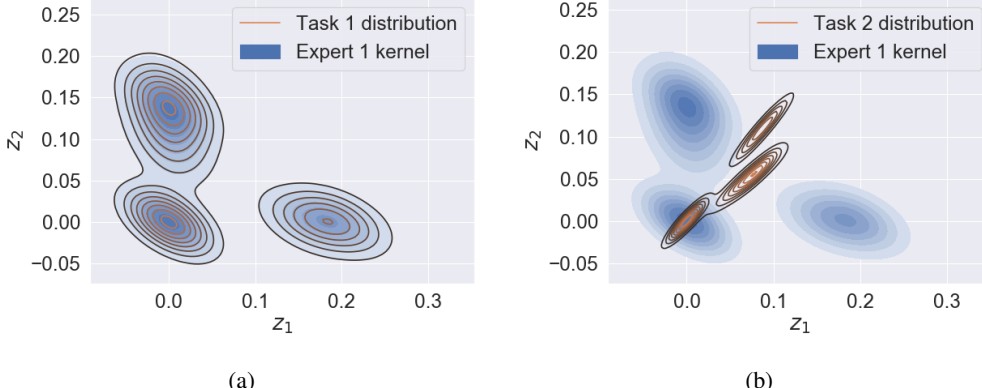

(a)             (b)

Figure 4: Difference between the extracted samples distribution and the kernels in the probabilistic layers. We use the hidden layers of the first expert to extract the features of the samples from task 1 and 2 respectively. The task likelihood is high if the two distributions are overlapped (e.g. Figure 4a, where samples are from Task 1, kernels are from: expert 1). The task likelihood will be decreased if they are different (e.g. Figure 4b, where samples are from Task 2, kernels are from: expert 1).

therefore the number of parameters used can be reduced. As shown in Figure 5a, the number of parameters used in TCNN is competitive to other methods after adding 20 experts.

Furthermore, Rusu et al. (2016); Yoon et al. (2017) re-use the neurons which have been trained with previous tasks. TCNN can be integrated with these techniques to save the resource. For example, we weighted sum the output features extracted by the convolutional layers of the previously learned expert, and add it to the current extracted features progressively (Rusu et al. (2016)). Shown in Equation 7, where $f_i(x)$ is the feature extracted by the $i_{th}$ expert, T is number of tasks, $\alpha_i$ is trainable weight parameter for $i_{th}$ expert, $\hat{f}_T(x)$ is the final extracted features of expert $T$. Besides of that, we can also share the parameters of the learned experts to the new experts directly. In this way, we do not need to train the convolutional neural networks again but train a weighted vector $\alpha$ instead, shown in Equation 8. We show the results in the Figure 5b, where the TCNN_progressive train the expert progressively by Equation 7, the TCNN_share is we train the last expert by the Equation 8. After combining with the techniques to reuse the exisiting neuron resources, the performance of the models do not degrade and similar to the baseline TCNN. The results show that the TCNN can integrate with these techniques to use the neuron resources efficiently and reduce the number of parameters for the future experts.

$$\hat{f}_T(x) = f_T(x) + \sum_{i=1}^{T-1} \alpha_i * f_i(x) \tag{7}$$

$$\hat{f}_T(x) = \sum_{i=1}^{T-1} \alpha_i * f_i(x) \tag{8}$$

Besides the parameter explosion, the inference time of TCNN increases with respect to the number of tasks. During the inference phase, we have to iterate all the learned experts to give the final prediction. One possible solution for this is early existing paradigm. Instead of running the experts in parallel, we put them in sequential manner. An expert will give a prediction if its confidence above threshold, or it passes the data to the next expert. We leave the improvement of TCNN and more comprehensive experiments as future work.

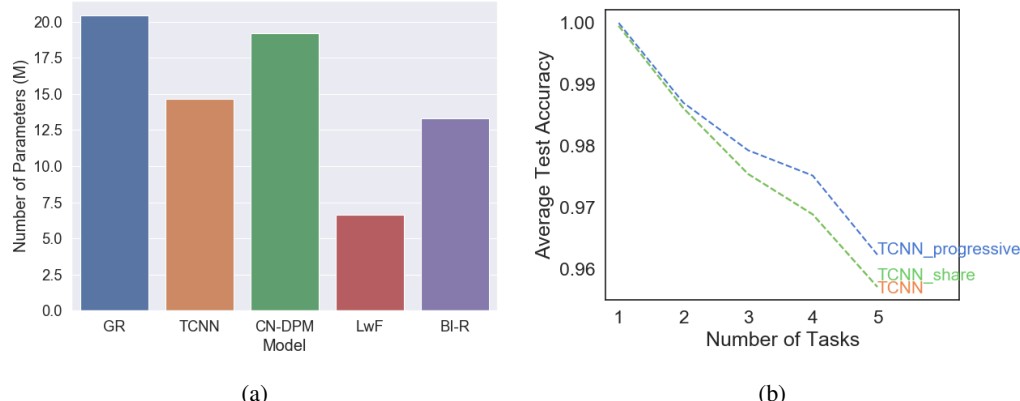

(a)             (b)

Figure 5: (a). Number of parameters used in CIFAR100 . (b). Supplementary split mnist experiments for reducing complexity of the TCNN. The TCNN_progressive and TCNN_share are trained by the Equation 7 and 8 respectively. Overall, by using progressive training and parameter sharing techniques, the performance of the model does not degrade. The accuracy fluctuates around the baseline TCNN model.

## 6 CONCLUSION

In this paper, we first discuss the reasons for forgetting problem in machine learning when different tasks are given to a model at different times. We demonstrate how providing or acquiring the learning task information is essential to address this issue. We also present a challenge that we have in our healthcare monitoring research and discuss how an automated and scalable model can solve this issue in dynamic and changing environments. We then propose a Task Conditional Neural Network (TCNN) model for continual learning of sequential tasks. TCNN is a novel expandable model that utilises a probabilistic layer to estimate the task likelihood.

TCNN consists of several different experts corresponding to different tasks. The model can learn and decide which expert should be chosen and activated under different tasks that are given to the model, without having provided the task information in advance. TCNN can detect the changes in the tasks and learn new tasks automatically. The proposed model implements these features by using a probabilistic layer and measuring the task likelihood given a set of augmented samples associated with a specific task. Our proposed model outperforms the state-of-the-art methods in terms of accuracy and the ability to learn the new tasks automatically. We hope this work can inspire more studies on investigating continual learning methods under dynamic environments.The future work will focus on transferring the knowledge between the experts, improving the task identification process, and defining dynamic and adaptive activation functions.

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
