# OpenReview forum: "Continual Learning Using Task Conditional Neural Networks"
_ICLR.cc/2022/Conference — ICLR 2022 Submitted_

### Official Review · Reviewer_sgG4 · 2021-10-31

**Correctness:** 3
**Technical Novelty And Significance:** 2
**Empirical Novelty And Significance:** 2
**Recommendation:** 5
**Confidence:** 4

**Main Review:**

Pros:
1. The paper is well-organized, easy to follow.
2. The proposed method takes advantage of Mixture of Experts (MoE), which can use a shallower network for each expert. This saves memory and computational cost.
3. The proposed method learns task-specific kernels to remember their data patterns for inferring the task identity, which is a new idea.

Cons:
It is difficult to infer task identity without directly accessing the old data distributions, and the proposed idea of learning task-specific kernels might be an effective solution. My concern is mainly on the empirical side:
1. The evaluation is on small-scale images, i.e. split-MNIST and split-CIFAR100. The evaluation of more complex domains with a larger scale is necessary since it will be more difficult to learn the data patterns and infer the task identities.
2. Replay-based methods are generally more advantageous in class-incremental scenarios since they directly recover old data distributions. However, the authors mainly compare with generative replay approaches. It would be helpful to compare with episodic replay approaches, such as iCaRL (CVPR17) and RWalk-S (ECCV18), and feature replay, such as REMIND (ECCV20) and Adam-NSCL (CVPR21).
3. The authors claim memory and computation efficiency as a key advantage of their method. However, since the entire model is expanding in continual learning, a comparison of memory and computational cost with other baselines should be provided.

Post rebuttal:
After reading the reply, the author has done partial experiments that show the effectiveness of the method. However, there still needs further experiments that may require a new review. Thus, I keep my score.

**Summary Of The Paper:**

This paper provides a new method to infer the task identity without directly accessing the old data distributions. The proposed method can learn task-specific experts with task-specific kernels to decide which expert should be chosen and activated under different tasks given to the model. The proposed method achieves strong performance on split-MNIST and split-CIFAR100 without requiring task information in advance.

**Summary Of The Review:**

Applying the MOE to continual learning is a new idea, but it should prove that inferring the task-specific identity has noticeable advantages over other memory buffer-based methods that are listed in the previous section, at least from the numerical perspective.

---

### Official Review · Reviewer_YEsG · 2021-11-01

**Correctness:** 3
**Technical Novelty And Significance:** 2
**Empirical Novelty And Significance:** 2
**Recommendation:** 3
**Confidence:** 4

**Main Review:**

Strengths:
The main idea is easy to understand. Task-incremental learning has been proven a much easier problem than class-incremental learning. Thus, a mechanism to predict the task identity will simplify the problem a lot.

Weakness:
1. It would be good to have a visualization to illustrate the high-level idea of the proposed method.
2. The novelty of the method is limited. The idea of task predictor has been shown in [1].
3. Catastrophic forgetting has always been a main issue in continual learning literatures. However, in this paper, how the proposed method contributes to alleviate forgetting is not clear. Moreover, is there a replay buffer to save previous examples?
4. The motivation of (1)(2) is somehow weak. Why a MoE-like formulation is good for modeling task likelihood? We can have a MLP with a softmax activation to model task likelihood as well. For example, a task classifier in [1] works also quite well.
5. For the experiments, it is not clear of how to set the augmentation functions and the parameters of the architecture is missing.
6. Also, an ablation study will be very helpful to understand the method. For example, changing the number of experts per task, changing augmentation functions, etc.

[1] Abati et al. Conditional Channel Gated Networks for Task-Aware Continual Learning. CVPR 2020.

**Summary Of The Paper:**

This paper attempts to proposed a new method called Task Continual Neural Network to address the problem of task identity inference in continual learning. The proposed method estimates the task likelihood by constructing a probabilistic layer based on the idea of mixture of experts (MoE).  Experiments on benchmark datasets demonstrate the effectiveness of this method.



**Summary Of The Review:**

Overall, I think the novelty of the method is limited, and critical insights and analysis are not enough, both theoretically and experimentally.

---

> ### Author Response · Authors · 2021-11-20
> **Response to Reviewer YEsG**
>
> We thank the reviewer for the time and comments. We address your comments below:
>
> 1. Different from [1], there is no task predictor and buffer for storing the previously learned tasks in our model. Different from them, each expert in our model can infer the task identity automatically and we do not need to access the previously learned samples anymore. The main advantage of this approach is the model will not bias to the samples in the buffer during the inference.
>
> 2. Similar to [1, 2], the model alleviate catastrophic forgetting by choosing different actions while encountering different tasks during the test state. Different from them, our model neither need to train a separate model to infer the task identity from buffer [1] nor to be informed of the task information during the test state [2]. The model will choose to fire the corresponding neurons by calculating the task likelihood.
>
> 3. We tried to make the settings as close to the real world as possible and assume the potential number of tasks to be learned is infinite. By using the MoE, we can keep adding new experts to the model while encountering new tasks. An MLP task predictor with a softmax actuation is limited in the following aspects:
> * The task number has to be defined in advance. Since the number of outputs is fixed, we have to initialise a different model every time while a new task coming or we know the number of tasks the model will encounter in future.
> * We need to train the task predictor while a new task is coming in.
> * The task predictor may bias to the samples in the buffer. While training a task predictor, we cannot store all the samples, but a small portion of the samples in a task. Hence the buffer size significantly affects the performance of the task predictor.
>
> 4. Thanks for the suggestions. We have added some missing content. We also add an ablation study in the Section 5.1.
>
>
> [1] Abati et al. Conditional Channel Gated Networks for Task-Aware Continual Learning. CVPR 2020.
>
> [2] Andrei A Rusu, Neil C Rabinowitz, Guillaume Desjardins, Hubert Soyer, James Kirkpatrick, Koray Kavukcuoglu, Razvan Pascanu, and Raia Hadsell. Progressive neural networks. arXiv preprint arXiv:1606.04671, 2016.

---

### Official Review · Reviewer_NcCF · 2021-11-02

**Correctness:** 2
**Technical Novelty And Significance:** 2
**Empirical Novelty And Significance:** 2
**Recommendation:** 3
**Confidence:** 4

**Main Review:**

It does make sense to try to use the MoE approach to handle continual learning (CL). The idea of learning task assignment probability is also reasonable to handle the task-agnostic CL problem. However, there are some problems with other motivations and approaches.
1. Although the proposed method proposes to learn task-specific kernel/anchor to predict the task for a sample, it is unclear how the model will be able to identify the new tasks (i.e., varying of the tasks) during training. If the data from different tasks come in the data stream without known boundaries, I do not see obvious evidence and design in the model to learn the boundaries of the tasks.
If the exact task number K has to be given, it will be a very strong restriction for a model to handle the task-agnostic CL setting.
2. It is not clear whether the model can explicitly identify the task boundaries via learning the task probability. If not, the approaches in (5) seem an ensemble of the MoE models training with different augmentation, and the task probability prediction works as the “heuristic attention”, which limits the novelty and significance of the work.
3. Experiments on larger datasets (at least like tiny Imagenet) should be conducted.

If I understand correctly, the code link on page 4 should not be included.


**Summary Of The Paper:**

This paper studies the task-agnostic continual learning problem by introducing a model called task conditional neural networks. The proposed method relies on the mixture of expert (MoE) approach to handle dynamically varying tasks and introduces “probabilistic layers” to predict the task index/assignment probability for each sample.

**Summary Of The Review:**

The basic ideas are well motivated, reasonable, and interesting. However, the motivation about how to approach the objective is improper, and the method is not carefully designed (or not completed). Experiments and analysis can be further improved.

---

> ### Author Response · Authors · 2021-11-20
> **Response to Reviewer NcCF**
>
> We thank the reviewer for the time and comments. We address your comments below:
>
> 1. The proposed model does not need task number K in advance and grows automatically while learning new tasks. The model keeps adding new while encountering new tasks.
>
> 2. In section 5.1, we add an experiment while the model sequentially learns different tasks without being informed of the task boundaries in the training state. The model turns into test mode after it is converged on the current task, and turns into training mode if it is unconfident with the encountering task. We assume the tasks are coming in sequence. To further analyse the performance of the model, we test the model on the previously learned tasks after learning a new one. The tasks are coming in the following manner: the training set of Task 1, the test set of Task 1, the training set of Task 2, the test set of Task 1 and 2 ... . In this setting, the model will add experts unnecessarily if it is unconfident with learned tasks, or fail to create new expert if it is confident with unlearned task. The results show that the model can identify the task boundaries while learning a stream of tasks.
>
> 3. We thanks the reviewer's suggestion. We will conduct a more complex experiment in our future work. Meanwhile, we think the current experiments can show the effectiveness of our model compared to the state-of-the-art.

---

### Official Review · Reviewer_H38t · 2021-11-02

**Correctness:** 3
**Technical Novelty And Significance:** 2
**Empirical Novelty And Significance:** 2
**Recommendation:** 3
**Confidence:** 4

**Main Review:**

The experiments do not report any indication of the size of the models for the different methods (e.g., number of parameters). The proposed method requires at least one expert per task and is therefore computational inefficient compared with other baseline.

On the subject of baselines, the baselines reported in this work are weak. It is unclear to me why some baselines were not reported despite being cited in the paper. For example,
- Andrei A Rusu, Neil C Rabinowitz, Guillaume Desjardins, Hubert Soyer, James Kirkpatrick, Koray Kavukcuoglu, Razvan Pascanu, and Raia Hadsell. Progressive neural networks. arXiv preprint arXiv:1606.04671, 2016.
- Joan Serrà, Dídac Surís, Marius Miron, and Alexandros Karatzoglou. Overcoming catastrophic forgetting with hard attention to the task. arXiv preprint arXiv:1801.01423, 2018.

The results reported (in Table 2) are not competitive with the state-of-the-art by some margin.

**Summary Of The Paper:**

This paper introduces Task Conditional Neural Network for continual learning. The method is based on mixture of experts. Each expert is independent of other experts and therefore the model is not susceptible to catastrophic forgetting.

**Summary Of The Review:**

Nice idea but weak experimental baselines and results.

---

### Decision · Program_Chairs · 2022-01-20

**Decision:**

Reject

**Comment:**

This paper proposes an expansion strategy for both task agnostic and task-boundary aware CL. The authors demonstrate the quality of their method using two-standard scenarios with the Split-MNIST and CIFAR datasets.

Enabling CL for task-agnostic and task-boundary aware is important and an active area of research. The proposed approach is an interesting method that adds an expert for each new task. Experts are then combined (Mixture of Experts) for prediction. One disadvantage of a MoE approach is that the model size and compute will grow linearly with the number of tasks. This effect is partly limited in the paper as the authors show that experts can be small neural networks.

There was a bit of confusion in the original reviews regarding the exact setting this paper works under. As far as I understand this paper mostly deals with the class-incremental setting (task IDs available at training time, but not at test time). The task agnostic setting (task IDs never given) is also explored in Section 5.1. I think this confusion is partly a reflection of the state of the CL literature and the authors provided clear and concise replies to the reviewers.

The main limitation that remains is regarding the experiments. I agree with the reviewers that the current experiments seem somewhat preliminary and showing results on larger scale datasets and/or compared to a wider diversity of baselines is important. Reviewer sgG4 made precise comments about this. Other minor comments by the reviewers including providing a detailed report of the memory usage and computational costs of the various methods (partly done in Figure 5.3).

I think this method is interesting and could be impactful. I strongly encourage the authors to polish their manuscript and consider adding some of the additional empirical results that were suggested.